# Accessing five oxidation states of uranium in a retained ligand framework

Chong Deng [1], Jiefeng Liang[1], Rong Sun[1,2], Yi Wang [1], Peng-Xiang Fu[1], Bing-Wu Wang[1,2], Song Gao [1,3] & Wenliang Huang [1] ✉

Understanding and exploiting the redox properties of uranium is of great importance because uranium has a wide range of possible oxidation states and holds great potential for small molecule activation and catalysis. However, it remains challenging to stabilise both low and high-valent uranium ions in a preserved ligand environment. Herein we report the synthesis and characterisation of a series of uranium(II–VI) complexes supported by a tripodal tris(amido)arene ligand. In addition, one- or two-electron redox transformations could be achieved with these compounds. Moreover, combined experimental and theoretical studies unveiled that the ambiphilic uranium–arene interactions are the key to balance the stabilisation of low and high-valent uranium, with the anchoring arene acting as a δ acceptor or a π donor. Our results reinforce the design strategy to incorporate metal–arene interactions in stabilising multiple oxidation states, and open up new avenues to explore the redox chemistry of uranium.

Uranium is the heaviest element abundant in nature. As an early actinide, uranium can exhibit multiple oxidation states and rich redox chemistry[1,2]. Understanding and exploiting the redox properties of uranium is not only pivotal for basic research[3], but also pressing for the nuclear industry[4–6], environmental sciences[7–9], and catalysis[10,11]. Five oxidation states, uranium(II) to uranium(VI), are well established, with a recent addition of a molecular uranium(I) complex[12]. Typically, low and high-valent uranium ions need a distinct coordination environment and thus may undergo substantial ligand rearrangement upon redox transformations[2,13,14]. For example, bis(trimethylsilyl)amide ([N(SiMe₃)₂]⁻) has been shown to form uranium(II–VI) complexes, whereas the lability of [N(SiMe₃)₂]⁻ results in a low structural rigidity and complicates the reaction outcomes under redox conditions[15–17]. On the other hand, chelating ligands with well-defined frameworks can provide a retained coordination environment, enabling a direct comparison of different oxidation states, and controllable redox transformations. Prominent examples include the tris(amido)amine [Tren$^{TIPS}$]³⁻, the tris(aryloxide)tris(amine) [($^{Ad}$ArO)₃tacn]³⁻, the bis(iminophosphorano)methanediide [BIPM$^{TMS}$]²⁻, and the tris(aryloxide)

arene [($^{Ad,Me}$ArO)₃mes]³⁻ (Fig. 1a). These chelating ligands made possible the isolation of the first terminal uranium nitride complex[18], a linear, O-coordinated η¹-CO₂ bound to uranium[19], an arene-bridged diuranium single-molecule magnet[20], and electrocatalytic water reduction[21], respectively. Remarkably, these achievements were made through redox processes, underlining the power of uranium redox chemistry within a retained ligand framework.

Despite great success, no chelating ligand has been shown to support all five well-established oxidation states of uranium, uranium(II–VI). For instance, while the electron-rich [Tren$^{TIPS}$]³⁻, [($^{Ad}$ArO)₃tacn]³⁻, and [BIPM$^{TMS}$]²⁻ have previously been shown to support uranium(III–VI)[22–25], the arene-anchored [($^{Ad,Me}$ArO)₃mes]³⁻ was found to stabilise uranium(II–V)[26,27]. We anticipated that the ambiphilic nature of arenes[28] might be utilized to balance the stability of low and high-valent uranium ions. It has been shown that weak π interactions exist between electrophilic uranium centres and neutral arenes[29–31], while uranium–arene δ interactions play a big role in inverse-sandwich uranium arene complexes[20,25,32–37], the stabilisation of unusual oxidation states[26,38,39], and the implementation of uranium

[1]Beijing National Laboratory for Molecular Sciences, College of Chemistry and Molecular Engineering, Peking University, Beijing 100871, P. R. China. [2]Beijing Key Laboratory for Magnetoelectric Materials and Devices, Beijing 100871, P. R. China. [3]Spin-X Institute, School of Chemistry and Chemical Engineering, State Key Laboratory of Luminescent Materials and Devices, Guangdong-Hong Kong-Macao Joint Laboratory of Optoelectronic and Magnetic Functional Materials, South China University of Technology, Guangzhou 510641, P. R. China. ✉e-mail: wlhuang@pku.edu.cn

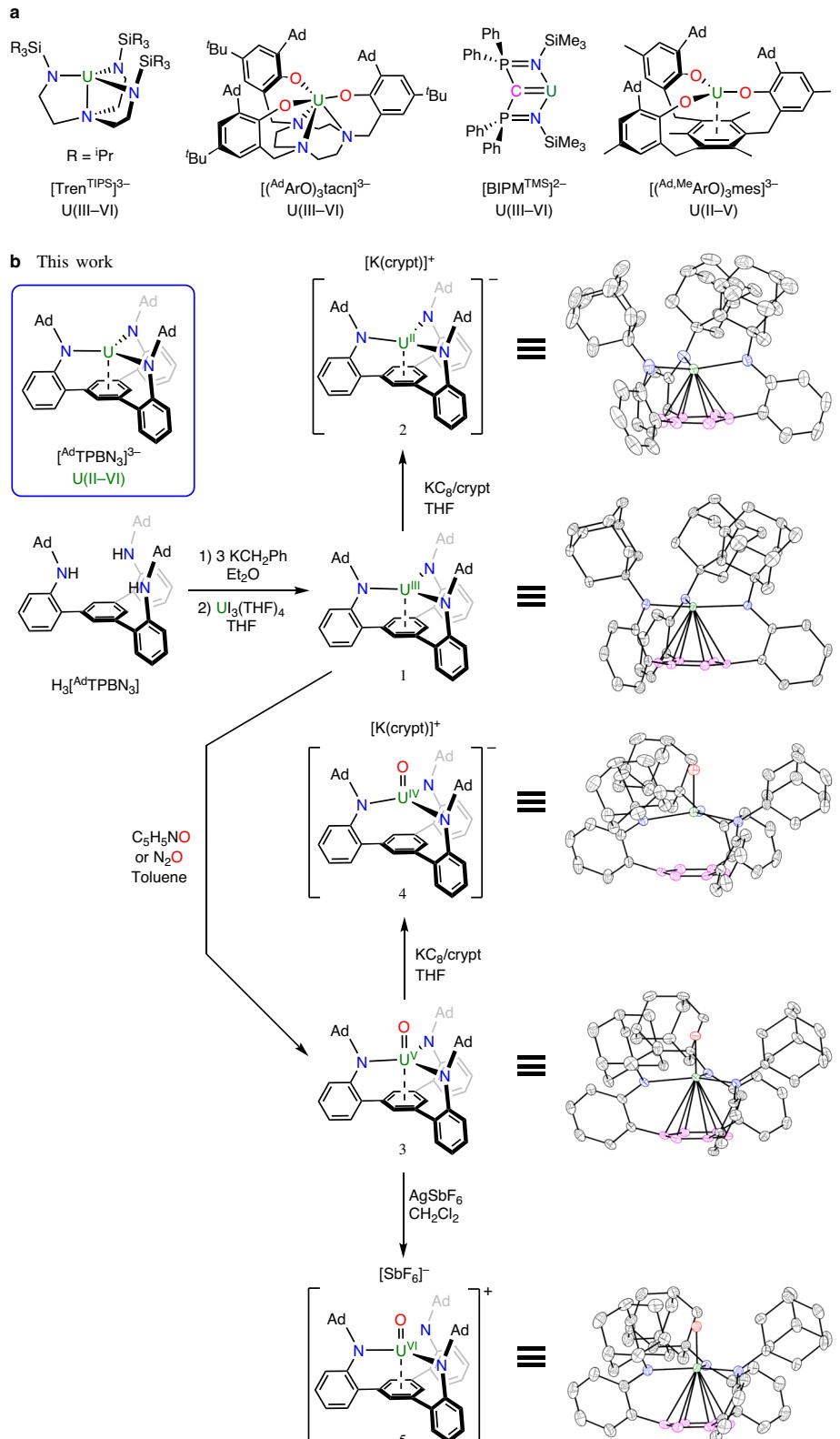

**Fig. 1 | Selected chelating ligands previously reported and synthesis and molecular structures in this work. a** Selected chelating ligands capable of supporting multiple oxidation states of uranium. **b** Synthesis and molecular structures of uranium(II–VI) complexes **1–5** supported by [$^{Ad}$TPBN$_3$]$^{3-}$. The single crystal structures are shown in thermal ellipsoids at 50% probability. All hydrogen atoms, counterions, and lattice solvents are omitted for clarity. Atom (colour): U (green), N (blue), O (red), C of the anchoring arene (pink), C of others (grey).

electrocatalysis[21,27]. Herein, we report the stabilisation of five oxidation states of uranium by a tris(amido)arene ligand through ambiphilic uranium–arene interactions, together with controlled redox transformations within this retained ligand framework.

## Results

### Synthesis and structural characterization

The pro-ligand 1,3,5-[2-(1-AdNH)$C_6H_4$]$_3C_6H_3$ (H$_3$[$^{Ad}$TPBN$_3$], Ad = 1-adamantyl) was prepared on a gram scale following a protocol similar to *N*-aryl tris(amido)arene pro-ligands[40]. In contrast to the fluxional behaviour of *N*-aryl tris(amido)arenes, H$_3$[$^{Ad}$TPBN$_3$] exhibits a $C_3$-syn structure with three nitrogen donors located at the same side of the 1,3,5-triphenylbenzene (TPB) backbone pointing inward (Supplementary Fig. 2). The pre-organized structure and improved crystallinity of [$^{Ad}$TPBN$_3$]$^{3-}$ provide ease for work-up and crystallization. Figure 1b illustrates the synthetic access to uranium(II–VI) complexes supported by [$^{Ad}$TPBN$_3$]$^{3-}$. Deprotonation of H$_3$[$^{Ad}$TPBN$_3$] by KCH$_2$Ph and subsequent salt metathesis with UI$_3$(THF)$_4$ yielded a uranium(III) complex ($^{Ad}$TPBN$_3$)U (**1**). Reduction of **1** by potassium graphite (KC$_8$) in the presence of 2,2,2-cryptand (crypt) in tetrahydrofuran (THF) generated a uranium(II) product [K(crypt)][($^{Ad}$TPBN$_3$)U] (**2**). On the other hand, oxidation of **1** by pyridine-*N*-oxide (C$_5$H$_5$NO) or N$_2$O in toluene furnished a uranium(V) terminal oxo complex ($^{Ad}$TPBN$_3$)UO (**3**). Furthermore, one-electron reduction or oxidation of **3** could be realized by KC$_8$/crypt in THF or silver hexafluoroantimonate (AgSbF$_6$) in dichloromethane (CH$_2$Cl$_2$), to afford the corresponding uranium(IV) and uranium(VI) terminal oxo complexes, [K(crypt)][($^{Ad}$TPBN$_3$)UO] (**4**) and [($^{Ad}$TPBN$_3$)UO][SbF$_6$] (**5**), respectively. These compounds were obtained in moderate to high yields and had good stability under inert atmosphere. For instance, no decomposition of the uranium(II) complex **2** in THF was observed even after prolong heating at 50 °C.

Compounds **1**–**5** were characterized by X-ray crystallography. The superpositions of the molecular structures (Fig. 2a) and key metrical parameters (Supplementary Table 1) of **1**–**5** reveal several features. Firstly, a ligand framework with a pseudo $C_3$ symmetry is retained for all compounds, with the decrease of the average U–N distances from 2.477(3) Å in **2** to 2.274(3) in **5** as the oxidation states of uranium increase. On the other hand, the U–C$_{centroid}$ distances vary considerably, which peak in **4** and decrease upon reduction or oxidation (Supplementary Fig. 13). **2** exhibits the shortest U–C$_{centroid}$ distance of 2.18 Å, and the average C–C distance of the anchoring arene in **2** (1.417(6) Å) is statistically not distinguishable from that in H$_3$[$^{Ad}$TPBN$_3$] (1.399(2) Å) by the 3σ-criterion, consistent with a uranium(II) ion

stabilised through δ backdonation[26,38,39]. **1** possesses a slightly longer U–C$_{centroid}$ distance of 2.34 Å, indicating weaker δ backdonation for the uranium(III) ion. For **3**–**5**, the U–C$_{centroid}$ distances decrease as the oxidation states of uranium increase, from 2.69 Å in **4** to 2.57 Å in **3**, and eventually to 2.49 Å in **5**. Notably, the U–C$_{centroid}$ distance in **3** is close to the U–C$_{centroid}$ distances of 2.546(1)–2.581(3) Å in π-bonded neutral arene complexes of uranium[29,30,41], but significantly shorter than the U–C$_{centroid}$ distances of 2.711(2) Å in another uranium(V) complex with an anchoring arene [((Ad,MeArO)$_3$mes)U(O)(THF)][27]. These structural features support our hypothesis that the anchoring arene may act as an additional ambiphilic ligand to balance the stabilisation of low-valent (II and III) and high-valent (V and VI) uranium ions. Furthermore, **3**–**5** represent the first trio of crystallographically authenticated uranium(IV–VI) terminal oxo complexes with the same supporting ligand. The U–O distances decrease as the oxidation states of uranium increase, from 1.874(4) Å in **4**, to 1.829(2) Å in **3**, and eventually to 1.818(2) Å in **5**, in line with literature values[23,42–44]. The U–O stretching frequencies of **3**–**5** obtained from the infrared (IR) spectroscopy (Supplementary Fig. 24) are within the literature range for uranium terminal oxo complexes[43,45–50]. Moreover, the trend of U–O stretching frequencies of **3**–**5** is in line with the trend of the U–O bond distances, indicating the U–O bond strength increases as the oxidation state of uranium increases in this trio.

### Electrochemistry and redox transformations

To further scrutinize the redox properties of these uranium complexes, electrochemical studies were carried out for **1** and **3**. The cyclic voltammogram of **1** revealed two one-electron events (Fig. 2b top). The reduction event at half-wave potentials $E_{1/2}$ = −2.40 V versus Fc$^+$/Fc (Fc = ferrocene) was assigned to a U$^{III}$/U$^{II}$ redox couple. According to the Randles-Ševčík analysis (Supplementary Fig. 31), the U$^{III}$/U$^{II}$ redox event is reversible at various scan rates between 20 and 800 mV/s. The reduction potential of **1** is less negative than that of [((Ad,MeArO)$_3$mes)U] (−2.495 V versus Fc$^+$/Fc)[51], in agreement with the higher stability of **2** than that of [K(crypt)][((Ad,MeArO)$_3$mes)U][26]. On the other hand, the oxidation event at $E_{1/2}$ = −0.12 V was assigned to a U$^{IV}$/U$^{III}$ couple, indicating chemically accessible one-electron oxidation of **1**. Indeed, treating **1** with excess silver fluoride or half an equivalent of 1,2-diiodoethane in toluene yielded ($^{Ad}$TPBN$_3$)UF (**6**) or ($^{Ad}$TPBN$_3$)UI (**7**), respectively. Intriguingly, further oxidation of **7** with silver nitrite resulted in the formation of **3**, probably through a ($^{Ad}$TPBN$_3$)U(ONO) intermediate[48], and the release of NO gas during the reaction was verified by the characteristic formation of Co(TPP)(NO)

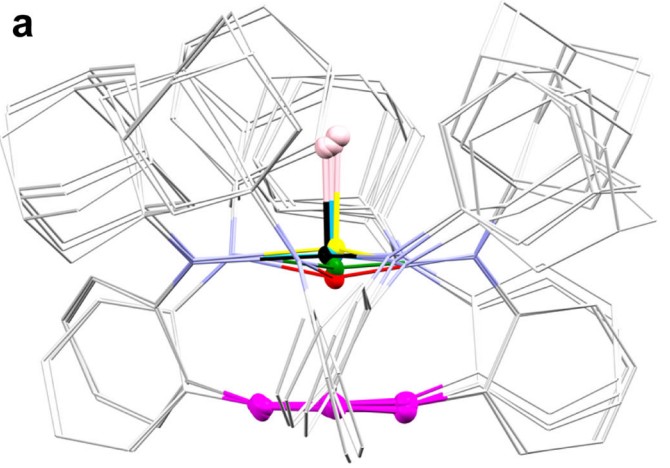
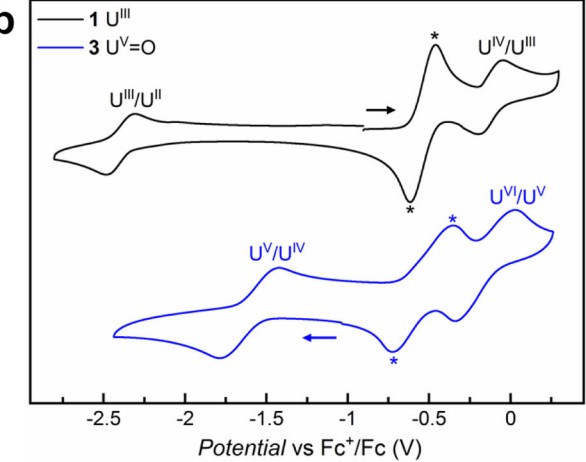

**Fig. 2 | Structural characterization and electrochemistry. a** Superpositions of molecular structures of **1**–**5**: C of the anchoring arene (magenta), O (pink), U(II) (red), U(III) (green), U(IV) (yellow), U(V) (sky blue), U(VI) (black). **b** Cyclic voltammograms of **1** (top) and **3** (bottom) at a scan rate of 200 mV/s in [$^n$Bu$_4$N][PF$_6$]/THF, with internal standard Fc$^{*+}$/Fc$^*$ (Fc$^*$ = decamethylferrocene) labelled with *.

(TPP = 5,10,15,20-teraphenylporphyrin) in the trapping experiment with Co(TPP) (see Supplementary Information section 1.3 for details). Moreover, **1** could also be converted to **4** by reacting with KNO$_2$ in the presence of crypt, representing a rare example of one-electron oxidation from uranium(III) to a uranium(IV) terminal oxo complex[52,53]. The cyclic voltammogram of **3** showed two reversible one-electron events at $E_{1/2}$ = −1.60 V and −0.16 V versus Fc$^+$/Fc, assigned to U$^V$/U$^{IV}$ and U$^{VI}$/U$^V$

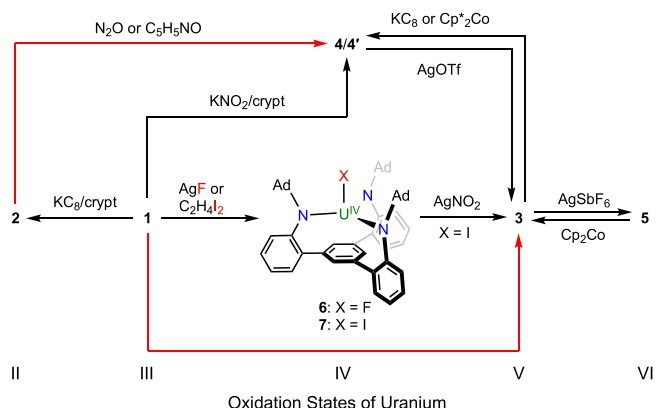

**Fig. 3 | Redox transformations for 1−7.** Black arrows for one-electron processes and red arrows for two-electron processes.

redox couples, respectively (Fig. 2b bottom). Based on these redox potentials, interconversions between **4** and **3** or **3** and **5** were realized by using appropriate oxidants (silver trifluoromethanesulfonate (AgOTf) or AgSbF$_6$) or reductants (KC$_8$ or Cp$_2$Co, Cp = cyclopentadienyl). A variant of **4**, [Cp*$_2$Co][($^{Ad}$TPBN$_3$)UO] (**4′**), could also be obtained via reduction of **3** with a mild reductant Cp*$_2$Co (Cp* = pentamethylcyclopentadienyl). Furthermore, two-electron oxidation of **2** to **4** could be accomplished by C$_5$H$_5$NO or N$_2$O, expanding the underdeveloped multi-electron redox chemistry of uranium(II)[39,54]. A full picture of redox transformations of uranium ions within the retained ligand framework of [$^{Ad}$TPBN$_3$]$^{3−}$ is illustrated in Fig. 3.

## Spectroscopic and magnetic studies
Various spectroscopic characterizations were performed to elucidate the electronic structures of this series of uranium complexes. The $^1$H NMR spectrum of **2** shows an upfield resonance at −72 ppm assigned to the protons of the anchoring arene (Supplementary Fig. 49), characteristic for uranium−arene δ interaction[33]. On the contrary, **5** exhibits a deshielded resonance at 9.13 ppm for the corresponding arene protons (Supplementary Fig. 56), indicating π-donation from the anchoring arene to the uranium(VI) ion. The UV−Vis−NIR spectra were recorded in THF for compounds **1−7** (Fig. 4a). Notably, the absorption spectrum of **2** has broad and intense bands in the visible and near-infrared regions, which is similar to other uranium(II) complexes with a $5f^46d^0$ electronic configuration[26,38], but different from $5f^36d^1$ uranium(II) ions[55]. The ligand-to-metal charge transfer bands are mostly in

**Fig. 4 | Spectroscopic and magnetic studies and theoretical calculations. a** UV−Vis−NIR spectra of **1−7** with the inset showing the NIR region. **b** Magnetic moments as a function of temperature (2−298 K) for uranium(II−V) complexes. **c** X-band EPR spectra of **1** in toluene and **2** in THF at 10 K. **d** Kohn-Sham orbitals (isosurface = 0.05) of the four SOMOs of **2**; hydrogens were omitted for clarity.

the ultra-violet region for **4**, but bathochromically shifted to visible and near-infrared regions for **3** and **5**. While **3** and **4** exhibit characteristic $f$–$f$ transitions in the near-infrared region for uranium(V) and uranium(IV) ions, respectively, **5** absorbs strongly over the entire range from 260–1600 nm with the absence of $f$–$f$ transitions, analogous to other uranium(VI) terminal oxo complexes[43,44]. Since X-ray photoelectron spectroscopy (XPS) has been shown to diagnostically identify the change of oxidation states for $f$-elements[56], we obtained XPS spectra for **1**–**7** (Supplementary Figs. 81–87). The binding energies of uranium 4$f$ orbitals show an increasing trend, as the oxidation states of uranium increase (Supplementary Table 6).

The electronic structures of these uranium complexes were further probed by superconducting quantum interference device (SQUID) magnetometry. Variable-temperature direct-current magnetic susceptibility data were collected under an applied magnetic field of 1 kOe for all compounds but diamagnetic **5** in solid state. The effective magnetic moments ($\mu_{eff}$) as a function of temperature are shown in Fig. 4b. The $\mu_{eff}$ of **2** at 298 K is 4.02 $\mu_B$, much higher than any previously reported uranium(II) complexes (2.2–2.8 $\mu_B$ at 300 K)[26,38,39,57,58] and the theoretical value of 2.68 $\mu_B$ for a free 5$f^4$ ion with a $^5I_4$ ground state. The high $\mu_{eff}$ of **2** may be attributed to the population of thermally accessible excited states, such as $^5I_5$ with a theoretical value of 4.93 $\mu_B$, because of strong bonding interactions between uranium and the anchoring arene. Upon lowering the temperature, the magnetic moments of **2** drop rapidly toward zero ($\mu_{eff}$ = 0.59 $\mu_B$ at 2 K), in line with other uranium(II) complexes[26,38,57,58]. While the temperature profiles of **1** and **3** are typical for uranium(III) and uranium(V), the high and low temperature magnetic moments of **4** ($\mu_{eff}$ = 4.53 $\mu_B$ at 298 K, and 2.58 $\mu_B$ at 2 K) are far exceeding the normal range of uranium(IV) ions[59]. Actually, to the best of our knowledge, both values of **4** are the highest in the literature for any single uranium(IV) ion. The presence of the strong axial oxo ligand and crystallographically three-fold symmetry may cause this anomalous magnetic behaviour of **4**, as shown in a recent magnetic study on $(UO[N(SiMe_3)_2]_3)^-$[60]. Notably, **6** also exhibits unusually large magnetic moments at low temperature ($\mu_{eff}$ = 1.59 $\mu_B$ at 2 K), while **7** behave normally ($\mu_{eff}$ = 0.53 $\mu_B$ at 2 K). The decrease of low temperature magnetic moments for **4**, **6**, and **7** corresponds with the descending bond strengths of the axial oxo, fluoro, and iodo ligands. Furthermore, the X-band electron paramagnetic resonance (EPR) spectra for **1**–**3** were collected at 10 K (Fig. 4c and Supplementary Fig. 97). The EPR spectra of **1** and **3** exhibited well-resolved anisotropic, nearly axial signals, as expected for uranium(III) and uranium(V) complexes with approximate $C_{3v}$ symmetry[26,61]. Simulation of the EPR spectra gave $g_{\parallel}$ value of 1.22 and $g_{\perp}$ values of 1.98 and 2.07 for **1**, and $g_{\parallel}$ value of 1.33 and $g_{\perp}$ values of 0.57 and 0.57 for **3**, consistent with $^4I_{9/2}$ ion and $^2F_{5/2}$ ion with a magnetic doublet ground state, respectively. The EPR spectrum of **2** only showed a small signal at $g$ = 2.00, which might be attributed to a radical impurity or solvated electrons ($g$ = 2.0023). The absence of EPR response for **2** is consistent with the assignment of a 5$f^4$ electronic configuration[26,38].

### Theoretical calculations

Density functional theory (DFT) calculations with scalar relativistic effects were performed on the full structures of **1**–**7** to probe their electronic structures, and in particular, the role of the anchoring arene in stabilising different oxidation states of uranium. The optimized structures match well with crystal structures. For **1**, three singly occupied molecular orbitals (SOMOs) are mainly composed of uranium 5$f$ orbitals, as expected for a uranium(III) ion. Among them, SOMO and SOMO−1 have minor contributions from the anchoring arene, indicating weak δ backdonation from uranium to the arene (Supplementary Table 14 and Figure 98). The δ backdonation is more prominent in the SOMOs 251α and 250α of **2**, featuring strong bonding interactions between uranium 5$f$ orbitals and the π* orbitals of the anchoring arene,

while the other two SOMOs, 249α and 248α, are predominantly uranium 5$f$ orbitals (Fig. 4d). The calculation results are consistent with experimental evidences, supporting the description that **2** is a 5$f^4$ uranium(II) complex stabilised through δ backbonding with the anchoring arene. For **3**–**5**, the composition analysis shows that while the uranium(IV) ion has few π interactions with the anchoring arene, the uranium(V) and uranium(VI) ions have appreciable π interactions with the anchoring arene (Supplementary Tables 16–18 and Figs. 100–102). These results are consistent with the elongated U–C$_{centroid}$ distance in **4** than **3** and **5**. Notably, the uranium–arene interactions have some mixing with U–O interactions in **3** and **5**, indicating possible inverse-trans-influence[62–64]. The natural localized molecular orbital (NLMO) analysis on U–O interactions shows one σ bond and two π bonds with increasing covalent character from uranium(IV) to uranium(VI) (Supplementary Tables 19–21 and Figs. 103–105), in line with the shortening of U–O distances. The multiple bonding character of U–O bonds is also supported by the Wiberg bond indexes, ranging from 1.52 to 1.93 (Supplementary Table 24). Other population analysis, including Mulliken atomic charges, spin populations, natural charges, and natural spin density (Supplementary Tables 25–28), gave similar pictures on the electronic structures of this series of uranium(II–VI) complexes.

To get a deeper insight on the uranium–arene interactions and their role in stabilising both low and high-valent uranium ions, extended transition state–natural orbitals for chemical valence (ETS–NOCV) calculations[65] were carried out for **1**–**5**, which were fragmented into the anchoring arene and the rest of the molecule. The σ, π, and δ-type uranium–arene interactions are calculated based on the symmetry of the NOCV pairs (Supplementary Tables 29–34). While the absolute values of stabilisation energies depend on the fragmentation and thus are arbitrary, the trend within a series is indicative of the relative strength of uranium–arene interactions. δ backdonation from uranium 5f orbitals to π* orbitals of the anchoring arene dominates in **1** and **2**, whereas π donations from π orbitals of the anchoring arene to uranium-based orbitals gradually strengthen as the oxidation states of uranium increase. Overall, the trend of total stabilisation energies of uranium–arene interactions correlate well with the trend of U–C$_{centroid}$ distances for **1**–**5** (Supplementary Fig. 108). The ETS–NOCV analysis further confirms that the ambiphilic uranium–arene interactions play a significant part in stabilising both low and high-valent uranium ions.

## Discussion

To summarize, a series of uranium(II–VI) compounds supported by a tripodal tris(amido)arene ligand were synthesized and characterised. Controlled two-way redox transformations could be readily achieved within the retained ligand framework. The electronic structures of these uranium complexes were scrutinized by structural, spectroscopic and magnetic studies together with DFT calculations, unveiling that the ambiphilic uranium–arene interactions play a pivotal role in stabilizing various oxidation states of uranium. The anchoring arene acts primarily as a δ acceptor for low-valent uranium ions and possesses increasing π donor characters as the oxidation states of uranium increase. The tripodal tris(amido)arene ligand framework capable of supporting five oxidation states of uranium will be an excellent platform to explore the redox chemistry of uranium, from installing multiply-bonded ligands to small molecule activation. Furthermore, the ligand design strategy disclosed here may be extended to other metals for supporting multiple oxidation states and enabling controllable redox transformations.

## Data availability

The X-ray crystallographic coordinates for structures reported in this study have been deposited at the Cambridge Crystallographic Data Centre (CCDC), under deposition numbers 2245010–2245020. These data can be obtained free of charge from The Cambridge Crystallographic Data Centre via www.ccdc.cam.ac.uk/data_request/cif.

Additional experimental, spectroscopic, crystallographic, and computational data are included in the Supplementary Information file. All other data are available from the corresponding author on request. Source data are provided with this paper.

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

## Acknowledgements

We gratefully acknowledge the National Natural Science Foundation of China (Grant No. 22171008) and the National Basic Research Program of China (No. 2018YFA0306003). We thank Dr. Jie Su for help with X-ray crystallography and Drs. Hui Fu and Xiu Zhang for help with NMR spectroscopy. The authors thank Beijing National Laboratory for Molecular Sciences and Peking University for financial support. C.D. thanks Peking University-BHP Carbon and Climate Wei-Ming PhD Scholars (No. WM202202) for support.

## Author contributions

C.D. prepared and characterized the compounds. J.L. performed the theoretical calculations. R.S. and B.W. obtained and analysed the SQUID data. Y.W. and C.D. collected and analysed crystallographic and electrochemical data. P.F. and C.D. obtained the EPR data and performed simulations. B.W., S.G. and W.H. acquired fundings. W.H. supervised the study. C.D., J.L. and W.H. wrote the manuscript with input from all of the authors.

## Competing interests

The authors declare no competing interests.
