## [Peer Review File · Nature Communications]

Accessing Five Oxidation States of Uranium in a Retained Ligand FrameworkReviewers' Comments:

Reviewer #1:

Remarks to the Author:

Uranium complexes supported by chelating ligands have provided some of the most exciting recent developments in providing new structural and bonding motifs. An arene-supported uranium tris-aryloxy complex allowed the second U(II) complex to be reported as well as catalytic reactivity with water, and arene-bound uranium complexes more generally provided interesting single-molecule magnet behaviour and other interesting reactivity modes. Here Deng et al. report the synthesis and characterisation of a novel series of structurally similar uranium tris-amido arene complexes spanning formal oxidation states from +2 to +6, including a study of their redox chemistry.

The combination of characterisation data collected for the complexes herein is convincing of their bulk formulations being correct. However I request that the authors collect and supply IR spectra as standard. Overall the paper is excellent, with succinct and precise analysis of the data. Most of the issues I have pointed out below are with the amount of literature coverage and citation, which needs to improve to put these results in context with previous work better. However I found the novelty of the results at a standard commensurate with this journal, and I am confident as the authors have suggested that this ligand framework should provide opportunities for new forays into reactivity studies and the stabilization of novel bonding motifs. I therefore think that the paper is accepted, as long as the minor corrections provided are addressed in full.

Abstract: I'd recommend deleting the sentence "In addition, two-way redox transformations could be achieved with these compounds;" it doesn't add anything that isn't already obvious (different U oxidation states are accessed by redox processes) and it isn't factually correct for the U(II) and U(VI) examples. I also recommend deletion of "Our results pave the way to harness the full redox capacities of uranium", as this is essentially meaningless hyperbole that far overextends beyond the facts of the results presented here. What about U(0) and U(I) (and possible negative U oxidation states)? Arenes have been used to stabilize U in a wide range of oxidation states in the same ligand framework previously by others (the results of Meyer et al with tris(aryloxy)mesitylene ligands spring immediately to mind, including for exploiting the arene ligand as a variable electron sink, but there are other examples). The way that the results are presented in this abstract doesn't put them in the full context of the literature.

Introduction: A better context with the previous literature is provided here but some things are not precise:

1. The authors state that the only ligand that stabilises U(II-VI) is N(SiMe₃)₂, but substituted cyclopentadienyls have done this too (admittedly some of the substituents and the number of rings often changes between oxidation states but this isn't precisely what is written, and it is also remiss not to cover a more complete list of citations to cover the contributions that this ligand class have made to the redox chemistry of U beyond the most recent U(I) example presented). To be precise, the coordination spheres of the complexes herein vary by the presence or absence of an oxo ligand, so strictly speaking the same ligand framework isn't truly "retained" between different U oxidation states.

2. The authors select two tris-aryloxy chelates and one triamidoamine pincer ligand as stabilizing multiple U oxidation states. Whilst I appreciate that not every ligand system can be covered here, I found it odd that two tethered aryloxides were covered and not bis(iminophosphorano) methanediides (BIPM-SiMe₃ has been the most used of this ligand class over closely related sulfur analogues), which have been shown to support U(III-VI) – I'd suggest either removal of the tacn-based framework to make room in figure 1(a) or finding a way to squeeze an extra chemdraw in, and adding appropriate references.

3. Where the authors state that other ligands "can support" and "is able to stabilise" certain ranges of oxidation states these should be changed to "has previously been shown to" – it isn't that the ligands

are not able to do this, just that they have not been shown to do this yet.

4. There is not enough coverage of uranium arene complexes. I think it is important for the authors to add this coverage and incorporate into in the later discussion. Importantly, arenes are redox non-innocent, that is covalency between the arene and U by pi- and delta-bonding makes the arbitrary assignment of oxidation states not truly reflective of some electron sharing between metal and ligand. This is especially true for some inverse arene low oxidation state U complexes, where stating formal oxidation states has sometimes been considered a moot point.

Figure 1 – The caption needs to additionally state that solid state structures shown are from single crystal XRD, the thermal ellipsoid probability level, that H atoms, counterions and lattice solvent are not shown for clarity, and a colour key for the atoms.

EPR spectroscopy: There needs to be some discussion of the g values extracted for the U(III) and U(V) complexes and how these correlate with the molecular structures. The authors attribute the signal at $g = 2.00$ to solvated electrons. The free electron value is $g = 2.0023$. The signal that the authors observe is best assigned as a free radical impurity rather than solvated electrons.

Supporting information:

1. FT-IR spectra need to be supplied for the full spectral range (4000-400 cm^{-1}) for all compounds, and lists of peaks summarised in experimental sections. This is a literature standard and allows facile comparison of vibrational modes between compounds. Given that most paramagnetic compounds lack ^{13}C NMR spectra this is good compensatory data for C-X bond information, and in this case the arene vibrational modes could shift with changes in U-binding by oxidation state; U=O bonds could also be assigned in U(V) and U(VI) compounds.
2. linewidths should be provided for broad peaks in ^1H NMR spectra as FWHM values in Hz in the experimental summaries of each complex.
3. Fig S78 – some of the orbitals have been cut off by the figure labels.

Reviewer #2:

Remarks to the Author:

Please see attached pdf file

The present study, *NCOMMS-23-13743-T*, submitted by Wenliang Huang and coworkers, reports, as titled, the stabilization of five oxidation states of uranium by their newly developed, arene-anchored tris-amido ligand through ambiphilic uranium–arene interactions, together with controlled redox transformations within this preserved ligand framework.

All new compounds were characterized by ^1H and ^{13}C NMR spectroscopy (where applicable) and CHN combustion analysis; hence, bulk purity is established. In addition, molecular structures for all new complexes were established by single-crystal XRD analysis. The standard complex characterization includes UV/vis/NIR, X-band EPR, and XPS spectroscopy, as well as SQUID magnetization and electrochemical studies. The report is complemented by computational analyses (DFT calculations and ETS-NOCV analysis), confirming – as expected – that ambiphilic uranium-arene interactions play a significant part in stabilizing both low and high-valent uranium ions.

Overall, the analyses and data interpretations are impeccable! I particularly appreciate their reasoning for the anomalous magnetic properties and the X-ray photoelectron spectroscopy confirming the change of formal oxidation states in their series of complexes; very nicely done.

However, each individual result presented here is not so novel. As correctly mentioned, and cited, electron-rich chelates support uranium in oxidation states III-VI, and the arene-anchored tris-aryloxide stabilizes uranium(II-V). Similarly, uranium oxo complexes exist in oxidation states IV, V, and VI, but the “first trio of crystallographically authenticated uranium(IV-VI) terminal oxo complexes with the same supporting ligand” is presented here.

That said, it is the full story that is quite impressive and very well presented. Also, the chemistry is timely, of broad interest, and important to chemists working in the field of coordination chemistry, likely not only concerning the f-elements. The manuscript is carefully prepared, precise, concise, and very well-written. The comprehensive picture of the molecular and electronic structure drawn here is exceedingly appealing. With all the new and little details, this reviewer finds this study quite educational and, thus, expects this paper to become very well-cited. Accordingly, this reviewer strongly recommends this submission for publication in *Nature Communications* upon minor revisions and consideration of the suggestions mentioned below.

Minor comments:

1. It is certainly a matter of taste, but this reviewer *suggests* displaying the molecular structures in a different style. This reviewer appreciates the first coordination sphere shown in color but would appreciate seeing these atoms plotted in the classic ORTEP style (just like the other C atoms) since there is information in this form of representation.
2. When discussing “slightly elongated C-C distances of the anchoring carbene,” this reviewer would appreciate doing so based on the 3-sigma criterion.
3. It is unusual (and maybe confusing) to have an internal $\text{Fe}(\text{Cp}^*)_2$ standard while the cyclovoltammogram is referenced vs. the $\text{Fe}(\text{Cp})^{+/0}$ standard. Is that necessary?
4. More importantly, however, this reviewer would appreciate the individual redox processes being studied and analyzed by the Randles–Ševčík equation.

5. Regarding the oxidation with nitrite salts, is there any evidence for the proposed U(ONO) intermediate, such as the detection of NO gas (as would be the case for the reaction of **7** with KNO₂ to yield **3**)?
6. Finally, this reviewer thinks that (upon publication) Fig. 2b and Figs. 4a-d could benefit from slightly larger axes and label font sizes.

Reviewer #3:

Remarks to the Author:

Recommendation: Reject.

Manuscript "Accessing Five Oxidation States of Uranium in a Retained Ligand Framework" describes the synthesis and characterization of a series of Uranium(II-VI) complexes supported by a tripodal tris(amido)arene ligand. Moreover, redox transformations were also shown within this retained ligand framework. The characterization of all the complexes have been done thoroughly including XRD. The chemistry reported, although performed competently, is rather conventional. The high oxidation species in this work are not very special as the [U=O] is blocking the access to strongly Lewis acidic Uranium center in U(IV-VI) complexes. Isolation of high oxidation U(IV-VI) complexes without the oxo-moiety would have been a great achievement. Uranium(II-VI) complexes have been known for many years (ref 19-22) (including the incorporation of arene bridge), so in my opinion stabilizing oxidation states from II to VI is not a groundbreaking achievement which would warrant publication of their manuscript in Nature Communications. I would rather recommend authors resubmit this manuscript to a more suitable journal such as Chemical Science.

There are some comments that authors should consider prior to resubmission.

1. In the abstract, authors stated understanding redox properties of Uranium is of great importance because of catalysis as one of the reasons. Now a days scientists are trying to even move away from Palladium, Platinum (all the precious metals) because of toxicity, cost etc. Even if we neglect the mild radioactivity of depleted Uranium, Uranium is highly toxic and using it in catalysis is not very desirable. So, talking about advantages of Uranium catalysis is misleading and authors should remove the term "catalysis".

2. Authors can explain choosing 'Adamantyl substituents' instead of 'Dipp version' (Inorg. Chem. 2021, 60, 20, 15321-15329) which authors reported recently.

3. Why is the U-Ccentroid distance in 1 (Oxidation state: III; 2.34 Å) shorter than the U-Ccentroid distance in 4 (Oxidation state: IV; 2.69 Å)?

4. Out of 9 reported crystals, five are modelled by using the SQUEEZE and no further details regarding the justification of using SQUEEZE routine is given in the SI. I believe it could be done better if modelling of solvent is done properly rather than just SQUEEZE. This is lazy crystallography and could be done better. The proper XRD is even more important here as the manuscript goes into details regarding the U-arene centroid distances which could be strongly influenced by the crystal packing effects and the final model of the XRD data.

5. A detailed table for the XRD parameters and restraints should be given, which is very easily generated by OLEX2, the software the authors are already using for refinements.

Reviewer #1 (Remarks to the Author):

Uranium complexes supported by chelating ligands have provided some of the most exciting recent developments in providing new structural and bonding motifs. An arene-supported uranium tris-aryloxide complex allowed the second U(II) complex to be reported as well as catalytic reactivity with water, and arene-bound uranium complexes more generally provided interesting single-molecule magnet behaviour and other interesting reactivity modes. Here Deng et al. report the synthesis and characterisation of a novel series of structurally similar uranium tris-amido arene complexes spanning formal oxidation states from +2 to +6, including a study of their redox chemistry.

The combination of characterisation data collected for the complexes herein is convincing of their bulk formulations being correct. However I request that the authors collect and supply IR spectra as standard. Overall the paper is excellent, with succinct and precise analysis of the data. Most of the issues I have pointed out below are with the amount of literature coverage and citation, which needs to improve to put these results in context with previous work better. However I found the novelty of the results at a standard commensurate with this journal, and I am confident as the authors have suggested that this ligand framework should provide opportunities for new forays into reactivity studies and the stabilization of novel bonding motifs. I therefore that the paper is accepted, as long as the minor corrections provided are addressed in full.

Response: We thank the reviewer for positive comments on the significance, novelty and technical parts of this work and helpful suggestion about the manuscript. In the revised manuscript, we have fully addressed the reviewer's comments, including supplying IR spectra for all new compounds and improving literature coverage and citation. We hope the reviewer find the revised manuscript suitable for publication in *Nature Communications*.

Abstract: I'd recommend deleting the sentence "In addition, two-way redox transformations could be achieved with these compounds;" it doesn't add anything that isn't already obvious (different U oxidation states are accessed by redox processes) and it isn't factually correct for the U(II) and U(VI) examples. I also recommend deletion of "Our results pave the way to harness the full redox capacities of uranium", as this is essentially meaningless hyperbole that far overextends beyond the facts of the results presented here. What about U(0) and U(I) (and possible negative U oxidation states)? Arenes have been used to stabilize U in a wide range of oxidation states in the same ligand framework previously by others (the results of Meyer et al with tris(aryloxide)mesitylene ligands spring immediately to mind, including for exploiting the arene ligand as a variable electron sink, but there are other examples). The way that the results are presented in this abstract doesn't put them in the full context of the literature.

Response: We appreciate the reviewer's insightful comments and helpful suggestion. For the first suggestion, we changed the sentence "two-way redox transformations could be achieved with these compounds" to "one- or two-electron redox transformations could be achieved with these compounds" in the abstract to avoid possible confusion for U(II) and U(VI). For the second suggestion, we changed the sentence "Our results pave the way to harness the full redox capacities of uranium, and the design strategy to incorporate ambiphilic metal-arene interactions may be

extended to other metals.” to “Our results reinforce the design strategy to incorporate metal–arene interactions in stabilising multiple oxidation states, and open up new avenues to explore the redox chemistry of uranium.”.

Introduction: A better context with the previous literature is provided here but some things are not precise:

1. The authors state that the only ligand that stabilises U(II–VI) is N(SiMe₃)₂, but substituted cyclopentadienyls have done this too (admittedly some of the substituents and the number of rings often changes between oxidation states but this isn’t precisely what is written, and it is also remiss not to cover a more complete list of citations to cover the contributions that this ligand class have made to the redox chemistry of U beyond the most recent U(I) example presented). To be precise, the coordination spheres of the complexes herein vary by the presence or absence of an oxo ligand, so strictly speaking the same ligand framework isn’t truly “retained” between different U oxidation states.

Response: We appreciate the reviewer’s insightful comments and helpful suggestion. We totally agree with the reviewer that the coordination spheres of uranium does vary slightly with or without oxo ligand for high and low-valent uranium ions, while the major coordination environment largely stays for uranium(II) to uranium(VI) in our ligand framework. We also agree with the reviewer’s comment that it is not strictly precise to state that the only ligand that stabilizes U(II–VI) is [N(SiMe₃)₂][−]. Therefore, we took the reviewer’s suggestion and modified the statement “Typically, low and high-valent uranium ions need a distinct coordination environment and thus may undergo substantial ligand rearrangement upon redox transformations². So far, the only ligand known to form uranium(II–VI) complexes is bis(trimethylsilyl)amide ([N(SiMe₃)₂][−]). However, the lability of [N(SiMe₃)₂][−] results in a low structural rigidity and complicates the reaction outcomes under redox conditions^{13–15}.” to “Typically, low and high-valent uranium ions need a distinct coordination environment and thus may undergo substantial ligand rearrangement upon redox transformations^{2,13,14}. For example, bis(trimethylsilyl)amide ([N(SiMe₃)₂][−]) has been shown to form uranium(II–VI) complexes, whereas the lability of [N(SiMe₃)₂][−] results in a low structural rigidity and complicates the reaction outcomes under redox conditions^{15–17}.” In addition to changing the statement, we also took the reviewer’s advice to include two reviews about the uranium cyclopentadienyl chemistry (ref. 13, Ephritikhine, M. Recent Advances in Organoactinide Chemistry As Exemplified by Cyclopentadienyl Compounds. *Organometallics* **32**, 2464–2488 (2013); ref. 14, Gremillion, A. J. & Walensky, J. R. in *Comprehensive Organometallic Chemistry IV* (eds Gerard Parkin, Karsten Meyer, & Dermot O’hare) 185–247 (Elsevier, 2022)).

2. The authors select two tris-aryloxide chelates and one triamidoamine pincer ligand as stabilizing multiple U oxidation states. Whilst I appreciate that not every ligand system can be covered here, I found it odd that two tethered aryloxides were covered and not bis(iminophosphorano) methanediides (BIPM-SiMe₃ has been the most used of this ligand class over closely related sulfur analogues), which have been shown to support U(III–VI) – I’d suggest either removal of the tacn-based framework to make room in figure 1(a) or finding a way to squeeze an extra chemdraw in, and adding appropriate references.

Response: We appreciate the reviewer's helpful suggestion. As mentioned by the reviewer, it is not possible to cover every chelating ligand system in Figure 1a for introduction. In the initial submission, we chose two tris(aryloxy) and one tri(amido)amine ligands because we think they all feature tripodal geometry and thus are most relevant to the tris(amido)arene ligand presented in the current work. However, since the reviewer mentioned the bis(iminophosphorano)methanediide ligand, which has a pincer geometry, and indeed this ligand has generated fruitful chemistry for uranium, we now included this ligand framework in Figure 1a. In addition, we modified the sentences in the introduction part "Prominent examples include the tris(amido)amine [Tren^{TIPS}]³⁻, the tris(aryloxy)tris(amine) [(^{Ad}ArO)₃tacn]³⁻, and the tris(aryloxy)arene [(^{Ad,Me}ArO)₃mes]³⁻ (Fig. 1a). These chelating ligands made possible the isolation of the first terminal uranium nitride complex¹⁶, a linear, O-coordinated η¹-CO₂ bound to uranium¹⁷, and electrocatalytic water reduction¹⁸, respectively." to "Prominent examples include the tris(amido)amine [Tren^{TIPS}]³⁻, the tris(aryloxy)tris(amine) [(^{Ad}ArO)₃tacn]³⁻, the bis(iminophosphorano)methanediide [BIPM^{TMS}]²⁻, and the tris(aryloxy)arene [(^{Ad,Me}ArO)₃mes]³⁻ (Fig. 1a). These chelating ligands made possible the isolation of the first terminal uranium nitride complex¹⁸, a linear, O-coordinated η¹-CO₂ bound to uranium¹⁹, an arene-bridged diuranium single-molecule magnet²⁰, and electrocatalytic water reduction²¹, respectively.", as well as the sentence in the second paragraph "while the electron-rich [Tren^{TIPS}]³⁻ and [(^{Ad}ArO)₃tacn]³⁻ can support uranium(III–VI)^{19,20}, the arene-anchored [(^{Ad,Me}ArO)₃mes]³⁻ is able to stabilise uranium(II–V)^{21,22}," changed to "while the electron-rich [Tren^{TIPS}]³⁻, [(^{Ad}ArO)₃tacn]³⁻, and [BIPM^{TMS}]²⁻ have previously been shown to support uranium(III–VI)²²⁻²⁵, the arene-anchored [(^{Ad,Me}ArO)₃mes]³⁻ was found to stabilise uranium(II–V)^{26,27}." Representative papers on uranium chemistry supported by [BIPM^{TMS}]²⁻ ligand is also cited in the revised manuscript: ref. 20, Mills, D. P., Liddle, S. T., et al. *Nat. Chem.* **3**, 454-460 (2011); ref. 24, Mills, D. P., Liddle, S. T., et al. *J. Am. Chem. Soc.* **134**, 10047-10054 (2012); ref. 25, Wooles, A. J., Liddle, S. T., et al. *Nat. Commun.* **9**, 2097 (2018).

3. Where the authors state that other ligands "can support" and "is able to stabilise" certain ranges of oxidation states these should be changed to "has previously been shown to" – it isn't that the ligands are not able to do this, just that they have not been shown to do this yet.

Response: We appreciate the reviewer's insightful comments and helpful suggestion. We agree with the reviewer and as mentioned in the previous response, we have changed the statement "while the electron-rich [Tren^{TIPS}]³⁻ and [(^{Ad}ArO)₃tacn]³⁻ can support uranium(III–VI)^{19,20}, the arene-anchored [(^{Ad,Me}ArO)₃mes]³⁻ is able to stabilise uranium(II–V)^{21,22}," to "while the electron-rich [Tren^{TIPS}]³⁻, [(^{Ad}ArO)₃tacn]³⁻, and [BIPM^{TMS}]²⁻ have previously been shown to support uranium(III–VI)²²⁻²⁵, the arene-anchored [(^{Ad,Me}ArO)₃mes]³⁻ was found to stabilise uranium(II–V)^{26,27}."

4. There is not enough coverage of uranium arene complexes. I think it is important for the authors to add this coverage and incorporate into in the later discussion. Importantly, arenes are redox non-innocent, that is covalency between the arene and U by pi- and delta-bonding makes the arbitrary assignment of oxidation states not truly reflective of some electron sharing between metal and ligand. This is especially true for some inverse arene low oxidation state U complexes, where stating formal oxidation states has sometimes been considered a moot point.

Response: We appreciate the reviewer's insightful comments and helpful suggestion. In the initial submission, we focused our discussion on the comparison with Meyer's tris(aryloxy) uranium complexes, since this system also features uranium-arene interactions and is thus most relevant to the current work. We agree with the reviewer that uranium arene complexes are of great importance, and therefore we have already cited some seminal papers in this field, including: Cesari, M., et al. *Inorg. Chim. Acta* **5**, 439-444 (1971); Cotton, F. A., et al. *Organometallics* **4**, 942-943 (1985); Emslie, D. J. H., et al. *Chem. Sci.* **13**, 13748-13763 (2022) for π -bound uranium neutral arene complexes, and Cummins, C. C., et al. *J. Am. Chem. Soc.* **122**, 6108-6109 (2000) and Liddle's comprehensive review (Liddle, S. T. Inverted sandwich arene complexes of uranium. *Coord. Chem. Rev.* **293-294**, 211-227 (2015)) for the δ -bonding inverse-sandwich arene complexes of uranium. In order to elaborate on this topic as suggested by the reviewer, in the revised manuscript, we added several more representative works discussing the structure and bonding of δ -bound uranium arene complexes following the sentence "while uranium-arene δ interactions play a big role in inverse-sandwich uranium arene complexes^{20,25,32-37}". Two of them utilizing the BIPM^{TMS} ligand have been introduced before and cited again here (ref. 20: Mills, D. P., Liddle, S. T., et al. *Nat. Chem.* **3**, 454-460 (2011); ref. 25: Wooles, A. J., Liddle, S. T., et al. *Nat. Commun.* **9**, 2097 (2018)). Additional references are included in the revised manuscript, including: ref. 34 Evans, W. J., Kaltsoyannis, N., et al. *J. Am. Chem. Soc.* **126**, 14533-14547 (2004); ref. 35 Patel, D., Liddle, S. T., et al. *Angew. Chem. Int. Ed.* **50**, 10388-10392 (2011); ref. 36 Arnold, P. L., et al. *Nat. Chem.* **4**, 668-674 (2012); ref. 37 Mougél, V., Mazzanti, M.; et al. *Angew. Chem. Int. Ed.* **51**, 12280-12284 (2012).

We would like to point out the major difference of this study from previous works in employing uranium-arene interactions. Previously, either δ interactions or π interactions is utilized for supporting electron-rich or electron-deficient uranium centres. In the current work, the tris(amido)arene ligand [^{Ad}TPBN₃]³⁻ can stabilise both low and high-valent uranium ions through ambiphilic uranium-arene interactions, i.e., both δ and π interactions, in a single system. This is also why we cover literatures of both π -bound uranium neutral arene complexes and δ -bound uranium reduced arene complexes. We also agree with the reviewer's statement that covalency in δ bonding interactions may complicate the assignment of oxidation states in some cases. In order to make a clear assignment of oxidation states of uranium, we compared our results with the literature precedence of Meyer (ref 26, La Pierre, H. S., Scheurer, A., Heinemann, F. W., Hieringer, W. & Meyer, K. Synthesis and Characterization of a Uranium(II) Monoarene Complex Supported by δ Backbonding. *Angew. Chem. Int. Ed.* **53**, 7158-7162 (2014).), Odom (ref 38, Billow, B. S. et al. Synthesis and Characterization of a Neutral U(II) Arene Sandwich Complex. *J. Am. Chem. Soc.* **140**, 17369-17373 (2018).), and Arnold's work (ref 39, Straub, M. D. et al. A Uranium(II) Arene Complex That Acts as a Uranium(I) Synthone. *J. Am. Chem. Soc.* **143**, 19748-19760 (2021).). Based on similar magnetic, EPR, and UV-Vis-NIR data and DFT calculation results, the uranium(II) supported by [^{Ad}TPBN₃]³⁻ ligand reported in this work is best described as a 5f⁴ ion stabilised by δ back-bonding, in line with the literature precedents.

Figure 1 – The caption needs to additionally state that solid state structures shown are from single crystal XRD, the thermal ellipsoid probability level, that H atoms, counterions and lattice solvent are not shown for clarity, and a colour key for the atoms.

Response: We appreciate the reviewer's careful catch and helpful suggestion. In the revised manuscript, we added this information in the caption of Figure 1 as the following: "The single crystal structures are shown in thermal ellipsoids at 50% probability. All hydrogen atoms, counterions, and lattice solvents are omitted for clarity. Atom (colour): U (green), N (blue), O (red), C of the anchoring arene (pink), C of others (gray)."

EPR spectroscopy: There needs to be some discussion of the g values extracted for the U(III) and U(V) complexes and how these correlate with the molecular structures. The authors attribute the signal at g = 2.00 to solvated electrons. The free electron value is g = 2.0023. The signal that the authors observe is best assigned as a free radical impurity rather than solvated electrons.

Response: We appreciate the reviewer's insightful comments and helpful suggestion.

(1) As suggested by the reviewer, we added discussion about g values for the U(III) and U(V) complexes in the revised manuscript, by elaborating the statement "the EPR spectra of **1** and **3** exhibited well-resolved anisotropic signals, typical for uranium(III) and uranium(V) ions" to "The EPR spectra of **1** and **3** exhibited well-resolved anisotropic, nearly axial signals, as expected for uranium(III) and uranium(V) complexes with approximate C_{3v} symmetry^{26,61}. Simulation of the EPR spectra gave g_{\parallel} value of 1.22 and g_{\perp} values of 1.98 and 2.07 for **1**, and g_{\parallel} value of 1.33 and g_{\perp} values of 0.57 and 0.57 for **3**, consistent with $^4I_{9/2}$ ion and $^2F_{5/2}$ ion with a magnetic doublet ground state, respectively." In the initial submission, we already included ref. 26 (La Pierre, H. S., Meyer, K.; et al. *Angew. Chem. Int. Ed.* **53**, 7158-7162 (2014)), which reported the EPR spectrum of a uranium(III) tris(aryloxide) compound. In the revised manuscript, we added ref. 61 (Bart, S. C., Meyer, K., et al. *J. Am. Chem. Soc.* **130**, 12536-12546 (2008)), which discussed about the EPR features of (pseudo) C_{3v} -symmetric uranium(V)-oxo/-imido complexes. The simulation spectra and parameters were already shown in the Supplementary Information of the initial submission.

(2) We agree with the reviewer that it is not trivial to distinguish the signal of solvated electrons from that of the free radical impurity, due to the absence of hyperfine coupling by EPR spectroscopy. In order to be more rigorous, we changed the statement "the EPR spectrum of **2** only showed a small signal at g = 2.00, attributed to solvated electrons" to "The EPR spectrum of **2** only showed a small signal at g = 2.00, which might be attributed to a radical impurity or solvated electrons (g = 2.0023).", in the revised manuscript.

Supporting information:

1. FT-IR spectra need to be supplied for the full spectral range (4000-400 cm^{-1}) for all compounds, and lists of peaks summarised in experimental sections. This is a literature standard and allows facile comparison of vibrational modes between compounds. Given that most paramagnetic compounds lack ^{13}C NMR spectra this is good compensatory data for C-X bond information, and in this case the arene vibrational modes could shift with changes in U-binding by oxidation state; U=O bonds could also be assigned in U(V) and U(VI) compounds.

Response: We appreciate the reviewer's valuable suggestion. In the revised Supplementary Information, we provide FT-IR spectra (4000–400 cm^{-1}) (new section S3 in the Supporting Information, Figs. S13–S23), as well as lists of peaks for all compounds reported in this work. In

all spectra, the range of 1700–900 cm^{-1} was obscured by peaks from the backbone of the ($\text{Ad}^{\text{d}}\text{TPBN}_3$) $^{3-}$ ligand, including 1,3,5-triphenylbenzene skeleton and 1-adamantyl substituents. This made it difficult to draw any conclusion about the arene vibrational modes with the changes in uranium–arene bonding as the oxidation states of uranium changes. Gratifyingly, the U–O stretching frequencies of the uranium(IV–VI) oxo complexes could be identified below 900 cm^{-1} (see Fig. S23 for tentative $\tilde{\nu}(\text{U}=\text{O})$ assignments: $[\text{U}^{\text{VI}}\text{O}][\text{SbF}_6]$, 789 cm^{-1} ; U^{VO} , 783 cm^{-1} ; $[\text{K}(\text{crypt})][\text{U}^{\text{IV}}\text{O}]$, 750 cm^{-1} ; $[\text{Cp}^*_2\text{Co}][\text{U}^{\text{IV}}\text{O}]$, 743 cm^{-1}). Although these assignments are not confirmed by ^{18}O labelling studies, the assigned peaks agree well with literature values (to list some: 751–753 cm^{-1} for $\text{U}^{\text{V}/\text{VI}}\text{O}$ in Burns, C. J., et al. *J. Am. Chem. Soc.* **115**, 9840-9841 (1993) & **117**, 9448-9460 (1995); 760–765 cm^{-1} for $\text{U}^{\text{IV}}\text{O}$ in Andersen, R. A., et al. *Organometallics* **24**, 4251-4264 (2005); 859–882 cm^{-1} for $\text{U}^{\text{VI}}\text{O}$ in Schelter, E. J., et al. *J. Am. Chem. Soc.* **135**, 511-518 (2013); 910 cm^{-1} for U^{VO} in Liddle, S. T., et al. *Angew. Chem. Int. Ed.* **52**, 4921-4924 (2013); 741–748 cm^{-1} for U^{VO} and 826–829 cm^{-1} for $\text{U}^{\text{VI}}\text{O}$ in Meyer, K., et al. *J. Am. Chem. Soc.* **136**, 11980-11993 (2014); 739 cm^{-1} for $\text{U}^{\text{IV}}\text{O}$ in Meyer, K., et al. *Dalton Trans.* **48**, 10853-10864 (2019)).

The trend of U–O stretching frequencies obtained from IR spectra is consistent with that of the U–O bond distances obtained from single crystal XRD studies, indicating the U–O bond strength increases as the oxidation state of uranium increases. Furthermore, the strengths of U(VI)–O and U(V)–O bonds are relatively close, and both are significantly larger than that of U(IV)–O bonds. Therefore, we added some discussion about U–O stretching frequencies in the revised manuscript, next to the section of XRD bond lengths comparison about U(IV–VI) oxo complexes as the following: “The U–O stretching frequencies of **3–5** obtained from the infrared (IR) spectroscopy (Supplementary Figure S23) are within the literature range for uranium terminal oxo complexes^{43,45-50}. Moreover, the trend of U–O stretching frequencies of **3–5** is in line with the trend of the U–O bond distances, indicating the U–O bond strength increases as the oxidation state of uranium increases in this trio.” In addition to ref. 43 (Meyer, K., et al. *J. Am. Chem. Soc.* **136**, 11980-11993 (2014)) and ref. 48 (Schelter, E. J., et al. *J. Am. Chem. Soc.* **135**, 511-518 (2013)), which were already cited in the initial submission, we added five more references in the revised manuscript: refs. 45-47 & 49-50 (refs. 45 & 46, Burns, C. J., et al. *J. Am. Chem. Soc.* **115**, 9840-9841 (1993) & **117**, 9448-9460 (1995); ref. 47, Zi, G., Andersen, R. A., et al. *Organometallics* **24**, 4251-4264 (2005); ref. 49, Liddle, S. T., et al. *Angew. Chem. Int. Ed.* **52**, 4921-4924 (2013); ref. 50, Meyer, K., et al. *Dalton Trans.* **48**, 10853-10864 (2019)).

2. linewidths should be provided for broad peaks in ^1H NMR spectra as FWHM values in Hz in the experimental summaries of each complex.

Response: We appreciate the reviewer’s helpful suggestion. In the revised Supplementary Information, we provided FWHM values for all the ^1H NMR broad peaks in the experimental section as well as in the caption of each ^1H NMR spectrum.

3. Fig S78 – some of the orbitals have been cut off by the figure labels.

Response: We appreciate the reviewer’s good catch. In the revised Supplementary Information, we fixed this issue (Fig. S99).

Reviewer #2 (Remarks to the Author):

The present study, NCOMMS-23-13743-T, submitted by Wenliang Huang and coworkers, reports, as titled, the stabilization of five oxidation states of uranium by their newly developed, arene-anchored tris-amido ligand through ambiphilic uranium–arene interactions, together with controlled redox transformations within this preserved ligand framework.

All new compounds were characterized by ^1H and ^{13}C NMR spectroscopy (where applicable) and CHN combustion analysis; hence, bulk purity is established. In addition, molecular structures for all new complexes were established by single-crystal XRD analysis. The standard complex characterization includes UV/vis/NIR, X-band EPR, and XPS spectroscopy, as well as SQUID magnetization and electrochemical studies. The report is complemented by computational analyses (DFT calculations and ETS-NOCV analysis), confirming – as expected – that ambiphilic uranium–arene interactions play a significant part in stabilizing both low and high-valent uranium ions.

Overall, the analyses and data interpretations are impeccable! I particularly appreciate their reasoning for the anomalous magnetic properties and the X-ray photoelectron spectroscopy confirming the change of formal oxidation states in their series of complexes; very nicely done.

However, each individual result presented here is not so novel. As correctly mentioned, and cited, electron-rich chelates support uranium in oxidation states III–VI, and the arene-anchored tris-aryloxyde stabilizes uranium(II–V). Similarly, uranium oxo complexes exist in oxidation states IV, V, and VI, but the “first trio of crystallographically authenticated uranium(IV–VI) terminal oxo complexes with the same supporting ligand” is presented here.

*That said, it is the full story that is quite impressive and very well presented. Also, the chemistry is timely, of broad interest, and important to chemists working in the field of coordination chemistry, likely not only concerning the *f*-elements. The manuscript is carefully prepared, precise, concise, and very well-written. The comprehensive picture of the molecular and electronic structure drawn here is exceedingly appealing. With all the new and little details, this reviewer finds this study quite educational and, thus, expects this paper to become very well-cited. Accordingly, this reviewer strongly recommends this submission for publication in *Nature Communications* upon minor revisions and consideration of the suggestions mentioned below.*

Response: We thank the reviewer for the very nice and comprehensive summary and highlights on this work as well as the positive comments and valuable suggestion. In the following, we have fully addressed the reviewer’s comments to improve the manuscript. We hope the revised manuscript is suitable for publication in *Nature Communications*.

Minor comments:

*1. It is certainly a matter of taste, but this reviewer *suggests* displaying the molecular structures in a different style. This reviewer appreciates the first coordination sphere shown in color but would appreciate seeing these atoms plotted in the classic ORTEP style (just like the other C atoms) since there is information in this form of representation.*

Response: We appreciate the reviewer’s helpful suggestion. We totally agree with the reviewer that classic ORTEP plot can provide additional information about the molecular structures. Actually, in the initial submission, the molecular structures in Figure 1b were drawn as thermal ellipsoids at 50%

probability, including uranium and atoms of the first coordination sphere (despite they were plotted in solid colour). In order to avoid confusion and improve the display of molecular structures, in the revised manuscript, we took the reviewer's advice and changed the "solid colour" of uranium and atoms of the first coordination sphere to ORTEP style with "shaded colour" for all molecular structures in Figure 1b.

2. When discussing "slightly elongated C-C distances of the anchoring carbene," this reviewer would appreciate doing so based on the 3-sigma criterion.

Response: We appreciate the reviewer's helpful suggestion. While likely the average C-C distance of the anchoring arene in **2** is slightly longer than that in the pro-ligand, they are within the 3 σ -criterion and thus statistically not distinguishable based on current single XRD data. In the revised manuscript, we changed the statement "**2** exhibits the shortest U-C_{centroid} distance of 2.18 Å and slightly elongated C-C distances of the anchoring arene (avg. C-C_{arene} of 1.417(6) Å in **2** versus 1.399(2) Å in H₃[^{Ad}TPBN₃])" by "**2** exhibits the shortest U-C_{centroid} distance of 2.18 Å, and the average C-C distance of the anchoring arene in **2** (1.417(6) Å) is statistically not distinguishable from that in H₃[^{Ad}TPBN₃] (1.399(2) Å) by the 3 σ -criterion".

3. It is unusual (and maybe confusing) to have an internal Fe(Cp*)₂ standard while the cyclovoltammogram is referenced vs. the Fe(Cp)⁺⁰ standard. Is that necessary?

Response: We appreciate the reviewer's careful catch and valuable comments. Actually, the use of an internal Fe(Cp*)₂ is necessary for reporting reliable and comparable data in this study. Under the conditions of our cyclic voltammetry experiment the peaks of the Fe(Cp)⁺⁰ standard were overlapped with the peaks originated from the solvent (THF) oxidation, which made it difficult to precisely reference the redox events, such as the oxidative process of complexes **1** (U^{III}) and **3** (U^VO). Inspired by some literature precedents in the organo-f-element chemistry (Evans, W. J., Yang, J. Y., et al. *Chem. Sci.* **12**, 8501-8511 (2021); Evans, W. J., et al. *Dalton Trans.* **50**, 14384-14389 (2021)), we turn to use another well-known internal standard Fe(Cp*)₂ (Lay, P. A., et al. *J. Phys. Chem. B* **103**, 6713-6722 (1999)) and resolved the issue. However, since Fe(Cp)⁺⁰ standard is recommended by IUPAC and is the most widely used in the field of organic and organometallic chemistry (Connelly, N. G. and Geiger, W. E. *Chem. Rev.* **96**, 877-910 (1996)), we took an extra step to reference the internal Fe(Cp*)₂ standard to Fe(Cp)⁺⁰ under the conditions of our cyclic voltammetry experiment. We are fully aware that this extra step may bring in some additional error; however, we think this is the best way to obtain reliable data and make it comparable with literature values (e.g., Meyer's U(III)/U(II) potential of tris(aryloxide)-arene complexes: La Pierre, H. S., Meyer, K.; et al. *Angew. Chem. Int. Ed.* **53**, 7154-7157 (2014)) at the same time. Moreover, we hope in this way our electrochemical data may be more readily used by future relevant studies for comparison.

4. More importantly, however, this reviewer would appreciate the individual redox processes being studied and analyzed by the Randles-Ševčík equation.

Response: We appreciate the reviewer's valuable suggestion. We agree with the reviewer that the Randles-Ševčík analysis may provide additional information about the electrochemical processes.

Actually, we have done so but unintentionally left this analysis out in the initial submission. In the revised Supplementary Information, we now included the analysis of individual redox processes (4 sets in total: U^{III}/U^{II} , U^{IV}/U^{III} , $U^{VO}/U^{IV}O$, $U^{VI}O/U^{VO}$), as well as the Randles-Ševčík analysis. Each set of analysis includes a stacked cyclic voltammogram of the individual redox event at various scan rates, an electrochemical parameter table of the individual redox event, and the Randles-Ševčík plots of the positive and negative peak currents of the redox event. According to these analyses, a good to excellent reversibility has been established for the individual redox processes reported in this study. In the revised manuscript, we also add a statement: “According to the Randles-Ševčík analysis (Supplementary Figure S30), the U^{III}/U^{II} redox event is reversible at various scan rates between 20 and 800 mV/s.” The newly added data can be found in the Supplementary Information (Figs. S29–S32, S35–S38, Tables S2–S5).

5. Regarding the oxidation with nitrite salts, is there any evidence for the proposed $U(ONO)$ intermediate, such as the detection of NO gas (as would be the case for the reaction of **7** with KNO_2 to yield **3**)?

Response: We appreciate the reviewer’s insightful comments and valuable suggestion. For the oxidation of $(^{Ad}TPBN_3)UI$ (**7**) by $AgNO_2$ to afford $(^{Ad}TPBN_3)O$ (**3**), after work-up, we observed the formation of yellow AgI instead of Ag^0 . This phenomenon is consistent with the formation of a $[U(IV)-ONO]$ intermediate, which could release “NO” to afford the final product **3**. However, no gas evolution was directly observed due to the slow rate of (heterogeneous) reactions. During the revision, we took the reviewer’s suggestion and attempted to detect the possible release of NO gas. The newly added experimental results are detailed in the new section “S1.3 Trapping Experiment” in the revised Supplementary Information. Following we briefly describe the results.

By utilizing an “apparatus” consisting of two vials with different size, the gaseous NO generated from the reaction could be trapped by a THF solution of $Co(TPP)$ (cobalt(II) 5,10,15,20-tetraphenylporphyrin), which has excellent affinity for NO gas and been used as a NO gas detector in the literature (To list some precedence: Hoard, J. L., et al. *J. Am. Chem. Soc.* **95**, 8281-8288 (1973); Richter-Addo, G. B., Kadish, K. M., et al. *Inorg. Chem.* **35**, 6530-6538 (1996); Szymczak, N. H., et al. *Chem. Sci.* **6**, 3373-3377 (2015); Fout, A. R., et al. *Science* **354**, 741-743 (2015); Matson, E. M., et al. *Chem. Commun.* **56**, 555-558 (2020)). Gratifyingly, the formation of the NO-bound complex, $Co(TPP)(NO)$, is confirmed by 1H NMR spectroscopy and IR vibrational spectroscopy. Both the chemical shifts of the diamagnetic peaks of $Co(TPP)(NO)$ (Fig. S62) and the diagnostic N–O stretching peak of $Co(TPP)(NO)$ (Fig. S24) are consistent with literature values. Some relevant references are added in the revised Supplementary Information for comparison (Fout, A. R., et al. *Science* **354**, 741-743 (2016); Gilbertson, et al. *Chem. Commun.* **52**, 11016-11019 (2016); Lee, C.-M., et al. *J. Am. Chem. Soc.* **142**, 8649-8661 (2020); Zhang, S., et al. *J. Am. Chem. Soc.* **144**, 2867-2872 (2022)).

The results of this trapping experiment support the NO-release process, and are consistent with a mechanism involving the formation of a $[U-ONO]$ intermediate. In the revised manuscript, we add a sentence to include this new result: “Intriguingly, further oxidation of **7** with silver nitrite resulted in the formation of **3**, probably through a $(^{Ad}TPBN_3)U(ONO)$ intermediate⁴⁸, and the release of NO

gas during the reaction was verified by the characteristic formation of Co(TPP)(NO) (TPP = 5,10,15,20-terraphenylporphyrin) in the trapping experiment with Co(TPP) (see Supplementary Information S1.3 for details).”.

6. Finally, this reviewer thinks that (upon publication) Fig. 2b and Figs. 4a-d could benefit from slightly larger axes and label font sizes.

Response: We thank the reviewer for helpful suggestion. In the revised manuscript, we enlarged the axes and label font sized for Fig. 2b and Figs. 4a-d. During the revision, we found a typo in the caption of Fig. 2 and corrected it by changing “C of the anchoring arene (violet)” to “C of the anchoring arene (magenta)”.

Reviewer #3 (Remarks to the Author):

Recommendation: Reject.

Manuscript “Accessing Five Oxidation States of Uranium in a Retained Ligand Framework” describes the synthesis and characterization of a series of Uranium(II–VI) complexes supported by a tripodal tris(amido)arene ligand. Moreover, redox transformations were also shown within this retained ligand framework. The characterization of all the complexes have been done thoroughly including XRD. The chemistry reported, although performed competently, is rather conventional. The high oxidation species in this work are not very special as the [U=O] is blocking the access to strongly Lewis acidic Uranium center in U(IV–VI) complexes. Isolation of high oxidation U(IV–VI) complexes without the oxo-moiety would have been a great achievement. Uranium(II–VI) complexes have been known for many years (ref 19-22) (including the incorporation of arene bridge), so in my opinion stabilizing oxidation states from II to VI is not a groundbreaking achievement which would warrant publication of their manuscript in Nature Communications. I would rather recommend authors resubmit this manuscript to a more suitable journal such as Chemical Science.

Response: We appreciate the reviewer’s positive comments on the thoroughness and rigor of the chemistry reported in this manuscript. However, we do not agree with the reviewer’s assessment of the novelty and importance of this work. First of all, the reviewer stated that uranium(II–VI) complexes have been known for many years. This is true for uranium(III) to uranium(V) but not for uranium(II). The first molecular U(II) complex was discovered in the last decade (Evans, W. J., Furche, F., et al. *J. Am. Chem. Soc.* **135**, 13310-13313 (2013)), and until now only a handful of molecular uranium(II) complexes have been isolated and characterized with a limited scope for supporting ligands. More importantly, while “Uranium complexes supported by chelating ligands have provided some of the most exciting recent developments in providing new structural and bonding motifs” as stated by reviewer 1, no chelating ligand has been shown to support all five oxidation states (+2 to +6) of uranium. In this study, we developed a tripodal tris(amido)arene ligand and showed this ligand framework is capable of supporting five oxidation states of uranium, i.e., uranium(II) to uranium(VI). Moreover, combined experimental and computational studies demonstrated that ambiphilic uranium-arene interactions play a key role in balancing the stability of both low- and high-valent uranium ions, which has not been shown before. Furthermore,

controlled one- or two-electron redox chemistry has been achieved with these complexes, which opens up new avenues in exploring redox chemistry of uranium (As stated by reviewer 1 “*I am confident as the authors have suggested that this ligand framework should provide opportunities for new forays into reactivity studies and the stabilization of novel bonding motifs.*”). Regarding high-valent uranium oxo complexes, while uranyl is most common for uranium(VI) and uranium(V), it is not trivial to prepare terminal uranium mono-oxo complexes, especially for multiple oxidation states, due to the high tendency to form oxo-bridged species induced by the potent nucleophilicity of the terminal oxo ligand (Hayton, T. W. Metal–ligand multiple bonding in uranium: structure and reactivity. *Dalton Trans.* **39**, 1145-1158 (2010)). This study represents the first trio of crystallographically authenticated uranium(IV–VI) terminal oxo complexes with the same supporting ligand, and establish a clear trend of U–O bond strength based on X-ray crystallography, IR stretching frequencies, and DFT calculations. Finally, the strategy to stabilise multiple oxidation states and enable redox chemistry has implications beyond f-block metal chemistry, as stated by reviewer 2 “*chemists working in the field of coordination chemistry, likely not only concerning the f-elements.*”. Other than oxidation states and redox chemistry, our study also provides several important discoveries that should advance the field, including: (1) A series of $5f^2$ uranium(IV) complexes in a pseudo C_{3v} geometry exhibits unusual magnetic behaviour due to a doubly degenerate electronic ground states; (2) A formal “oxygen radical anion transfer” from KNO_2 to uranium(III) to generate uranium(IV) terminal oxo products is revealed for the first time; (3) Inverse-trans-influence observed in uranium(V) and uranium(VI) terminal oxo complexes with a trans arene donor, instead of traditional strong donors like N and O atoms.

Based on the above points and reasonings, we believe that in terms of novelty and importance, this work meets the standard for publication in *Nature Communications*, as recommended by reviewer 1 and 2. We hope reviewer 3 also find the revised manuscript suitable for publication in *Nature Communications*.

There are some comments that authors should consider prior to resubmission.

1. In the abstract, authors stated understanding redox properties of Uranium is of great importance because of catalysis as one of the reasons. Now a days scientists are trying to even move away from Palladium, Platinum (all the precious metals) because of toxicity, cost etc. Even if we neglect the mild radioactivity of depleted Uranium, Uranium is highly toxic and using it in catalysis is not very desirable. So, talking about advantages of Uranium catalysis is misleading and authors should remove the term “catalysis”.

Response: We thank the reviewer to raise up this point. We would like to take it as an opportunity to elaborate on this topic a bit more in the response. We agree with the reviewer that in certain cases, such as pharmaceutical production, chemists are trying to avoid using metal catalysts, especially those containing heavy and precious metals like palladium and platinum, because of their high cost and potential toxicity due to metal contaminations in the products. However, that is not saying metal catalysts are not valuable. Actually, palladium compounds have been used even more often in the initial stage of drug discovery, since it allows fast diversification of drug candidates (see M. R. Uehling et al., *Science* **363**, 405–408 (2019) as an example). Moreover, for large-scale industrial processes, whose products are not directly consumed by the human-beings, such as water splitting,

dinitrogen activation, and carbon dioxide reduction, the toxicity of metal catalyst is not a big issue. And this field is where uranium catalysis may find good use. Indeed, the great potential of uranium in catalysis was discovered as early as the beginning of the 20th century, when Haber's initial patent on ammonia production (Haber, F. Verfahren Zur Herstellung von Ammoniak Durch Katalytische Vereinigung von Stickstoff Und Wasserstoff, zweckmäßig Unter Hohem Druch. German patent DE229126, 1909) showed uranium-based catalysts has a high catalytic competency in the Haber-Bosch process. The development of uranium-based catalytic reactions has continued, with homogeneous uranium catalysis experienced a renaissance in the past two decades. The uranium-based catalysis has been comprehensively reviewed multiple times on prestigious journals, to list some: Cummins, C. C., et al. Towards uranium catalysts. *Nature* **455**, 341-349 (2008); Meyer, K., et al. From Chemical Curiosities and Trophy Molecules to Uranium-Based Catalysis: Developments for Uranium Catalysis as a New Facet in Molecular Uranium Chemistry. *JACS Au* **1**, 698-709 (2021); Arnold, P. L. and Turner, Z. R., Carbon oxygenate transformations by actinide compounds and catalysts. *Nat. Rev. Chem.* **1**, 0002 (2017); Gaunt, A. J. *Chem. Rev.* **113**, 1137-1198 (2013); Mathur, S., et al. *ACS Catal.* **9**, 4719-4741 (2019). In recent years, significant breakthroughs in uranium catalysis have been reported, exemplified by the electrocatalytic water reduction (Meyer, K., et al. *Nature* **530**, 317-321 (2016)) and N₂ fixation and functionalization (Arnold, P. L., et al. *Nat. Chem.* **12**, 654-659 (2020)), both utilizing well-defined chelating ligand frameworks capable of supporting multiple oxidation states of uranium. It is almost certain that more uranium mediated synthetic transformations and catalysis will be unveiled in the near future. Our work also provides a unique, promising platform to explore redox chemistry of uranium for potential catalytic transformations.

Moreover, developing uranium-based catalysis is not only promising, but also urgent. Depleted uranium is continuously generated in large quantities from the production and consumption of nuclear fuels (World Nuclear Association, The Nuclear Fuel Cycle (2021)) (not to mention military use). For example, in 2000, the USA's SNF (spent nuclear fuel with ²³⁸U as the major (>95 wt%) radionuclide) inventory was about 42,300 tHM (metric tons of heavy metal), increasing by around 2000 tHM every year, while the legislated capacity for the potential high-level waste repository at Yucca Mountain is 70,000 tHM of SNF equivalent (Hu, Q.-H., et al. *J. Environ. Radioact.* **101**, 426-437 (2010)). How to make good use of depleted uranium presents a significant challenge for chemists (Kaltsoyannis, N., Liddle, S. T. *Chem* **1**, 659-662 (2016)), since the long-time storage of the "uranium waste" in the form of UF₆ (a highly corrosive, volatile material) may cause serious problems on environmental and public health (Hon, Z., et al. *Sustainability* **7**, 4063-4077 (2015)). Developing uranium-based catalysis can turn this unwanted, environmental unfriendly "uranium waste" to high-value industrial materials, providing a sustainable long-term solution for the nuclear industry. In this sense, the cost of depleted uranium may be regarded as negative, since it will save a lot of spend on storing UF₆ waste as well as reduce the potential environmental hazard of depleted uranium.

In sum, based on the above reasonings, we believe it is appropriate to mention uranium catalysis in the Abstract and Introduction, since developing uranium catalysis is promising, urgent and relevant to this work.

2. Authors can explain choosing 'Adamantyl substituents' instead of 'Dipp version' (Inorg. Chem. 2021, 60, 20, 15321–15329) which authors reported recently.

Response: We appreciate reviewer's notice of our previous work on developing tripodal tris(amido)arene ligands. In the manuscript of the initial submission, we have already discussed about the choice of substituents and the reasonings in the first paragraph of Results and Discussion Section: "In contrast to the fluxional behaviour of *N*-aryl tris(amido)arenes, H₃[^{Ad}TPBN₃] exhibits a C₃-syn structure with three nitrogen donors located at the same side of the 1,3,5-triphenylbenzene (TPB) backbone pointing inward (Supplementary Figure S1). The pre-organized structure and improved crystallinity of [^{Ad}TPBN₃]³⁻ provide ease for work-up and crystallization." As clearly stated in the main text, we choose 'adamantyl substituents' instead of 'Dipp version', because of the pre-organized structure of the pro-ligand and improved crystallinity of the uranium(III) complexes supported by [^{Ad}TPBN₃]³⁻ ligand, which provides ease for work-up and characterization.

3. Why is the U–C_{centroid} distance in **1** (Oxidation state: III; 2.34 Å) shorter than the U–C_{centroid} distance in **4** (Oxidation state: IV; 2.69 Å)?

Response: As emphasized in the manuscript, the anchoring arene can serve as an ambiphilic ligand to support both low and high-valent uranium ions. For low-valent uranium ions, i.e., uranium(II) and uranium(III), the δ back-donation from uranium 5f orbitals to π* orbitals of the anchoring arene results in significant shortening of the U–C_{centroid} distance; on the other hand, for high-valent uranium ions, i.e., uranium(V) and uranium(VI), the π-donation from π orbitals of the anchoring arene to uranium-based orbitals also leads to shorter U–C_{centroid} distance. This leaves uranium(IV) as an outlier, since uranium(IV) in this ligand system is neither a good δ donor nor a good π acceptor (especially for the U–O moiety in **4**). Therefore the U–C_{centroid} distance in uranium(IV) complex **4** is the longest among all oxidation states. In the manuscript, we have discussed the trend of U–C_{centroid} distances as a whole, since it demonstrates the ambiphilic nature of uranium–arene interactions. Fig. 2a depicted the superpositions of molecular structures of **1–5**, featuring the relative positions of uranium ions within the ligand framework. We also discussed about this in the crystallographic section "For **3–5**, the U–C_{centroid} distances decrease as the oxidation states of uranium increase, from 2.69 Å in **4** to 2.57 Å in **3**, and eventually to 2.49 Å in **5**. Notably, the U–C_{centroid} distance in **3** is close to the U–C_{centroid} distances of 2.546(1)–2.581(3) Å in π-bonded neutral arene complexes of uranium^{29,30,41}, but significantly shorter than the U–C_{centroid} distances of 2.711(2) Å in another uranium(V) complex with an anchoring arene [((^{Ad,Me}ArO)₃mes)U(O)(THF)]²⁷. These structural features support our hypothesis that the anchoring arene may act as an additional ambiphilic ligand to balance the stabilisation of low-valent (II and III) and high-valent (V and VI) uranium ions."

In addition, we performed DFT calculations and ETS-NOCV analysis to reveal the uranium–arene interactions and discussed about it in the main text: "For **3–5**, the composition analysis shows that while the uranium(IV) ion has few π interactions with the anchoring arene, the uranium(V) and uranium(VI) ions have appreciable π interactions with the anchoring arene (Supplementary Tables S16–S18 and Figures S97–S99). These results are consistent with the elongated U–C_{centroid} distance in **4** than **3** and **5**." and "δ backdonation from uranium 5f orbitals to π* orbitals of the anchoring

arene dominates in **1** and **2**, whereas π donations from π orbitals of the anchoring arene to uranium-based orbitals gradually strengthen as the oxidation states of uranium increase. Overall, the trend of total stabilisation energies of uranium–arene interactions correlate well with the trend of U–C_{centroid} distances for **1–5** (Supplementary Figure S105).” In addition, in the Supplementary Information of the initial submission, we included Supplementary Table S29 and Fig. S105 to show the correlation between uranium oxidation states or U–C_{centroid} distances with the calculated stabilisation energies for compounds **1–5**. It can be drawn from these data that the total stabilisation energy of the uranium–arene interaction in **1** is significantly greater than that of **4**, consistent with the U–C_{centroid} distances determined by X-ray crystallography. With all these data, analysis and discussion, we believe that not only the reasoning for “U–C_{centroid} distance in **1** (Oxidation state: III; 2.34 Å) shorter than the U–C_{centroid} distance in **4**”, but also the trend of U–C_{centroid} distances and uranium–arene interactions for the whole series of uranium(II–VI) complexes have been rationalized clearly in the manuscript of our initial submission.

4. Out of 9 reported crystals, five are modelled by using the SQUEEZE and no further details regarding the justification of using SQUEEZE routine is given in the SI. I believe it could be done better if modelling of solvent is done properly rather than just SQUEEZE. This is lazy crystallography and could be done better. The proper XRD is even more important here as the manuscript goes into details regarding the U-arene centroid distances which could be strongly influenced by the crystal packing effects and the final model of the XRD data.

Response: We appreciate the reviewer’s insightful comments and helpful suggestion. We totally agree with the reviewer about the importance of appropriate refinement, especially for the reliability of crystallographic data that are discussed in the manuscript. First of all, we apologize for a typo in the Supplementary Information in the initial submission. It was wrongfully stated that “... SQUEEZE was used in the structure refinement of **1**, **2**·0.5THF, **3**·C₇H₈, **4**’, and **6**”; actually, the refinement of **6** was done without SQUEEZE. Thus, we remove “**6**” from the sentence in the revised Supplementary Information. Secondly, after carefully checking our crystal data, we noticed that in the refinement of [Cp*₂Co][(^{Ad}TPBN₃)UO], an equivalent of lattice solvent molecule of THF was “omitted” by SQUEEZE. Hence, in the revised materials, we modelled this THF molecule and provided an updated version for [Cp*₂Co][(^{Ad}TPBN₃)UO]·THF (**4**’·THF) without the use of SQUEEZE. We carefully checked the Supplementary Information and revised all related contents. It is worth mention that the new version refined without SQUEEZE is comparable to the original one done with SQUEEZE in terms of key structural parameters (e.g., U–O distance, U–C_{arene} distances, U–3N_{plane} distance).

For other compounds (**1**, **2**·0.5THF, **3**·C₇H₈) modelled using SQUEEZE, we checked the original data and have tried our best to model the solvents without SQUEEZE. However, we found it was necessary to use SQUEEZE due to severe disorder or other limitation of the lattice solvents. As suggested by the reviewer, we show further details for each crystal refinement using SQUEEZE in **S2.2. Details of restraints and SQUEEZE** in the revised Supplementary Information. In the refinement of **2**·0.5THF and **3**·C₇H₈, the modelling of the solvent molecules could be done well, but the SQUEEZE routine is still necessary, reflecting that the co-crystallization of the solvent is a common phenomenon in our system due to slow crystallization processes in dilute solutions. Among

11 crystals reported in the manuscript (actually not 9), now only three of them (**1**, **2**·0.5THF, **3**·C₇H₈) are modelled using SQUEEZE, and this proportion is not abnormal in the fields of organometallic chemistry and coordination chemistry. Some recent publications for comparison: 2 out of 9 using SQUEEZE, in Figueroa, J. S., et al. *Science* **375**, 1393-1397 (2022); 12 out of 16, in Otten, E., et al. *J. Am. Chem. Soc.* **142**, 20170-20181 (2020); 2 out of 2, in Yam, V. W.-W., et al. *J. Am. Chem. Soc.* **142**, 2448-2459 (2020); 1 out of 1, in Sun, Z.-M., et al. *Nat. Commun.* **13**, 2149 (2022).) We believe that our crystallography is at an acceptable level and hope the reviewer find the crystal refinement after revision is reasonable.

Lastly, regarding the influence of the crystal packing effects on key structural parameters, in the current study, we think this influence is very small if any. For example, we have two different types of crystals for the U(V) terminal oxo complex **3**, one co-crystallized with an equivalent of toluene and the other without co-crystallizing solvent (Figs. S4 and S5). They share very similar structural parameters (Table S9). Therefore, we believe the discussion about key metrical parameters based on the SRD data is valid in the manuscript.

5. A detailed table for the XRD parameters and restraints should be given, which is very easily generated by OLEX2, the software the authors are already using for refinements.

Response: We thank the reviewer for helpful suggestion. In the initial submission, we provided XRD parameters in the caption of each figure of molecular structures in the Supplementary Information, including one paragraph about bond lengths, bond angles, torsion angles, and another paragraph about XRD parameters, including crystallographic data and structure refinement information. Because we have 11 crystal structures and a single table to include all the XRD parameters will be too large, so we prefer to list this information in the caption under each molecular structure. For restraints, as suggested by the reviewer, we provided all the restraints used for the refinement of single crystal data in **the newly added section S2.2. Details of restraints and SQUEEZE** in the revised Supplementary Information.

Reviewers' Comments:

Reviewer #1:

Remarks to the Author:

I thank the authors for their careful and considered response to the reviewers comments. I believe that all points made have been addressed by full in the revised submission and I have no further points to add. My congratulations go to the authors for producing an excellent paper.

Reviewer #2:

Remarks to the Author:

All of my previous concerns and suggestions have been adequately addressed. Accordingly, I recommend the publication of this rebuttal in its present form and congratulate the authors for this beautiful and significant work.

Reviewer #3:

Remarks to the Author:

A revised version of Nature Communications manuscript NCOMMS-23-13743A has been reviewed. The manuscript has been improved in the process of editing. While I am still not completely convinced that the manuscript describes sufficient novelty for publishing in Nature communications.

1. It is true that authors isolate the first trio of uranium(IV-VI) terminal oxo complexes with the same supporting ligand and all three complexes are structurally characterized. But uranium(IV-V) terminal oxo complexes reported earlier (ref. 27; Nat. Chem. 10, 259-267 (2018)) and uranium(V-VI) terminal oxo complexes also reported (ref. 23; J. Am. Chem. Soc. 134, 5284-5289 (2012)). The synthetic routes are also the same. As I previously pointed out, high oxidation state uranium(IV-VI) complexes without the oxo-moiety would have been a great achievement.

2. I agree that U(II) complexes are less explored than other uranium oxidation states and no chelating ligand has been shown to support all five oxidation states (+2 to +6) of uranium previously. But the previous reported arene supported ligand system [(Ad,MeArO)₃mes]₃⁻ stabilized four oxidation states (+2 to +5) (ref. 26-27). So, I think this is not a huge upgrade from the previous report. Due to those points, I have a reservation to accept this work in Nature Communications.

Reviewer #1 (Remarks to the Author):

I thank the authors for their careful and considered response to the reviewers comments. I believe that all points made have been addressed by full in the revised submission and I have no further points to add. My congratulations go to the authors for producing an excellent paper.

Response: We thank the reviewer for constructive comments and insightful advices that really help us improve the manuscript.

Reviewer #2 (Remarks to the Author):

All of my previous concerns and suggestions have been adequately addressed. Accordingly, I recommend the publication of this rebuttal in its present form and congratulate the authors for this beautiful and significant work.

Response: We thank the reviewer for constructive comments and insightful advices that really help us improve the manuscript.

Reviewer #3 (Remarks to the Author):

A revised version of Nature Communications manuscript NCOMMS-23-13743A has been reviewed. The manuscript has been improved in the process of editing.

Response: We thank the reviewer for constructive comments and insightful advices that really help us improve the manuscript.

While I am still not completely convinced that the manuscript describes sufficient novelty for publishing in Nature communications.

1. It is true that authors isolate the first trio of uranium(IV–VI) terminal oxo complexes with the same supporting ligand and all three complexes are structurally characterized. But uranium(IV-V) terminal oxo complexes reported earlier (ref. 27; Nat. Chem. 10, 259-267 (2018)) and uranium(V-VI) terminal oxo complexes also reported (ref. 23; J. Am. Chem. Soc. 134, 5284-5289 (2012)). The synthetic routes are also the same. As I previously pointed out, high oxidation state uranium(IV-VI) complexes without the oxo-moiety would have been a great achievement.

Response: Although several chelating ligands have been shown to support two oxidation states of the uranium–oxo linkage, it is still challenging to support terminal uranium oxo with multiple oxidation states within a similar ligand framework. Our work achieved the isolation of the first trio of uranium(IV–VI) terminal oxo complexes, which allowed us to establish the U–O bond strength with various characterization methods, including X-ray crystallography and IR stretching frequencies. Moreover, our work revealed that upon oxidation, the uranium centre become more electrophilic, which results in increased π donation from the anchoring arene. Besides, in addition

to traditional synthetic routes, we developed new pathways to synthesize uranium(IV) oxo from U(III) and KNO₂ and to synthesize uranium(V) oxo from U(IV) iodide and AgNO₂.

We agree with the reviewer that stabilizing high-valent uranium without oxo moiety would be a worthwhile synthetic target. However, this target is beyond the scope of the current study. Relevant projects are ongoing in our laboratory toward this end.

2. I agree that U(II) complexes are less explored than other uranium oxidation states and no chelating ligand has been shown to support all five oxidation states (+2 to +6) of uranium previously. But the previous reported arene supported ligand system [(^{Ad,Me}ArO)₃mes]³⁻ stabilized four oxidation states (+2 to +5) (ref. 26-27). So, I think this is not a huge upgrade from the previous report.

Response: Our work not only introduced a well-defined ligand framework capable of supporting uranium(II) to uranium(VI), but also demonstrated, by comprehensive experimental and computational studies, that the ambiphilic uranium–arene interactions play a key role in balancing the stability of both low- and high-valent uranium ions, which has not been shown before. This strategy to stabilise multiple oxidation states and enable redox chemistry has implications beyond f-block metal chemistry.

Due to those points, I have a reservation to accept this work in Nature Communications.

Response: We respect the reviewer's opinion. However, we are confident that the significance and novelty of this work warrant its publication in *Nature Communications*.